# Impact of Colistin Dosing on the Incidence of Nephrotoxicity in a Tertiary Care Hospital in Saudi Arabia

**DOI:** 10.3390/antibiotics9080485

**Published:** 2020-08-06

**Authors:** Reem Almutairy, Waad Aljrarri, Afnan Noor, Pansy Elsamadisi, Nour Shamas, Mohammad Qureshi, Sherine Ismail

**Affiliations:** 1Pharmaceutical Care Department, Ministry of National Guard Health Affairs, King Abdullah International Medical Research Center, King Saud Bin Abdulaziz University for Health Sciences, Jeddah 21423, Saudi Arabia; reem.rfm@gmail.com (R.A.); w.s.aljrarri@gmail.com (W.A.); afnan.m.noor@gmail.com (A.N.); 2Pharmacy Department, Beth Israel Deaconess Medical Center, Boston, MA 02115, USA; pelsamad@bidmc.harvard.edu; 3Infection Prevention and Control Department, Ministry of National Guard Health Affairs, King Abdullah International Medical Research Center, King Abdulaziz Medical City, Riyadh 11426, Saudi Arabia; shamasno@ngha.med.sa; 4University of Toronto, University Health Network, Division of Nephrology, Toronto, ON M5G 2N2, Canada; mohammadazfar.qureshi@uhn.ca

**Keywords:** colistin, colistin sulfate, polymyxins, colistin/administration and dosage, colistin/toxicity, acute kidney injury/drug effect, acute kidney injury/drug therapy, multi-drug resistance, multiple drug resistance

## Abstract

Colistin therapy is associated with the development of nephrotoxicity. We examined the incidence and risk factors of nephrotoxicity associated with colistin dosing. We included adult hospitalized patients who received intravenous (IV) colistin for >72 h between January 2014 and December 2015. The primary endpoint was the incidence of colistin-associated acute kidney injury (AKI). The secondary analyses were predictors of nephrotoxicity, proportions of patients inappropriately dosed with colistin according to the Food and Drug Administration (FDA), European Medicines Agency (EMA), and Garonzik formula and clinical cure rate. We enrolled 198 patients with a mean age of 55.67 ± 19.35 years, 62% were men, and 60% were infected with multidrug-resistant organisms. AKI occurred in 44.4% (95% CI: 37.4–51.7). Multivariable analysis demonstrated that daily colistin dose per body weight (kg) was associated with AKI (OR: 1.57, 95% CI: 1.08–2.30; *p* = 0.02). Other significant predictors included serum albumin level, body mass index (BMI), and severity of illness. None of the patients received loading doses, however FDA-recommended dosing was achieved in 70.2% and the clinical cure rate was 13%. The incidence of colistin-associated AKI is high. Daily colistin dose, BMI, serum albumin level, and severity of illness are independent predictors of nephrotoxicity.

## 1. Introduction

The emergence of multidrug-resistant (MDR) Gram-negative microorganisms is a worldwide health concern [1]. Infections involving these organisms are associated with delayed initiation of the appropriate antimicrobial therapy and increased health-care costs and mortality [2]. Colistin, an antibiotic discovered in the late 1940s and approved by the FDA in 1962, exhibits activity against a wide range of Gram-negative pathogens, including MDR organisms [3,4]. Although effective, its use was abandoned because of a high incidence of associated nephrotoxicity. Nevertheless, it has recently been reintroduced into practice because of the inadequacy of other antimicrobial agents against MDR organisms. Despite the fact that colistin has been available for over 50 years, the optimal dosing regimen has not been defined.

Colistin is formulated as the prodrug colistimethate sodium. Two main colistimethate sodium products are available: one approved by the FDA and the other by the EMA. In recent years, the pharmacokinetics and pharmacodynamics of colistin have been further elucidated [5]. This prompted the FDA and EMA to update their dosing recommendations in 2013 and 2014, respectively [3,6]. Although both updates were as a result of the same body of evidence, significant variation exists in their dosing recommendations. A recent study that compared both dosing regimens found that the EMA regimen, which includes higher overall maintenance doses, is more likely to achieve the target steady-state concentration of colistin than the FDA recommendations [4,7]. Additionally, only the EMA advocates the use of a loading dose as proposed by an earlier pharmacokinetic study in critically ill patients [8]. The study also provided a formula to calculate the optimal loading and maintenance doses of colistin based on serum colistin concentrations.

One of the most significant drawbacks of the use of colistin is the high incidence of associated nephrotoxicity. However, it remains controversial whether this adverse effect is dose-dependent, time-dependent, or both [9,10]. According to several studies, the incidence of colistin-induced nephrotoxicity ranges from 31 to 60% [11,12,13,14]. These studies evaluated different dosing regimens and found different rates of nephrotoxicity in their study populations. Several risk factors can increase a patient’s risk of developing colistin-induced nephrotoxicity, including chronic kidney disease, diabetes mellitus, cancer, anemia, and exposure to multiple events harmful to the kidneys (e.g., critical illness, sepsis, shock, burns, trauma, major surgery, and nephrotoxin exposure) [15,16]. However, colistin-induced nephrotoxicity is reversible and rarely results in permanent kidney damage [17,18]. The risk of nephrotoxicity must be weighed against the consequences of inadequately treating a life-threatening infection [17,18].

As there was no clear guidance for dosing colistin in our hospital and the presence of evolving updates in the dosing following Garonzik and EMA recommendations at the time we conducted the study, there was a lack of data on the development of colistin-associated AKI in our setting and its association with dosing. Given the high frequency of colistin use at our institution, we examined the incidence of colistin-induced nephrotoxicity in our patient population and evaluated the association between colistin dosing and the incidence of nephrotoxicity, predictors of colistin-associated nephrotoxicity and clinical cure rate associated with our current dosing regimen.

## 2. Materials and Methods

### 2.1. Study Design and Setting

We conducted a retrospective chart review study at the King Khalid Hospital, King Abdulaziz Medical City, Jeddah, Saudi Arabia. The Informatics Department generated a list of all patients who received IV colistin between January 2014 and December 2015.

### 2.2. Study Participants

Patients were included in the study if they were aged 18 years or older and had received IV colistin therapy for at least 72 h for treatment of infections as per the discretion of the treating physician. Patients were excluded if they had AKI at the time of initiation of colistin therapy, as defined by an increase in the serum creatinine concentration of ≥1.5 times of the baseline value as per Kidney Diseases Initiative Global Outcomes (KDIGO) guidelines for AKI [19]; lacked data on baseline serum creatinine concentration; required renal replacement therapy before colistin initiation; received only inhaled colistin; or were pregnant. For patients who received more than one course of colistin, only the first course was included. All data were collected using the electronic health records (EHRs) system and patients’ charts.

### 2.3. Study Outcomes

The primary outcome was the incidence of AKI, which was defined as an increase in serum creatinine concentration of ≥1.5 times the baseline value during the treatment course of colistin as defined by KDIGO [19]. Secondary analyses included the stage of AKI as defined by KDIGO [19], predictors of AKI, and impact of colistin dosing on the incidence of nephrotoxicity. Additionally, we assessed the proportions of patients inappropriately dosed according to dosing recommendations by the FDA [4], EMA [6], and Garonzik et al. [8] and determined the clinical cure rate associated with our current dosing regimen.

### 2.4. Data Collection

The data collected included baseline demographic information; colistin dosing and duration of therapy; concomitant antibiotic administration; and laboratory findings such as albumin, hemoglobin, white blood cell, and serum creatinine levels (evaluated at baseline, peak (if AKI occurred), and the end of colistin treatment). We also recorded isolates of the bacterial cultures and susceptibility patterns to various antimicrobial agents and defined MDR organisms for Enterobacterales, *A. baumannii* complex, and *P. aeruginosa* as the resistance to three or more classes of antimicrobials [20]. Patients who developed AKI were followed up for 3 months after the end of colistin treatment to evaluate its reversibility. To assess the severity of illness and risk of mortality in our patient population, we calculated the Charlson Comorbidity Index (CCI) [21,22] and Acute Physiologic Assessment and Chronic Health Evaluation (APACHE) II scores of ICU patients [22]. Additionally, we recorded concomitant use of nephrotoxic drugs such as vancomycin, amphotericin-B, aminoglycosides, non-steroidal anti-inflammatory drugs, angiotensin-converting enzyme inhibitors, angiotensin-II receptor blockers, contrast media, acyclovir, diuretics, and vasopressors. We evaluated the clinical cure rate of those who completed at least 7 days of colistin therapy, which we defined as the resolution of leukocytosis and fever within 48 h of day 7 of colistin therapy as per previous studies [10,23].

### 2.5. Sample Size

A sample of 194 patients was calculated to provide a 95% confidence limits of ± 7% around an assumed incidence of AKI of 45% [11,12,13,14,24].

### 2.6. Statistical Analysis

Descriptive statistics such as means ± standard deviation, medians, and interquartile ranges (IQRs) and percentages were used to present baseline demographics. Exact binomial calculations were used to estimate the incidence of AKI and 95% Confidence intervals (CIs). The chi-squared test was used to compare the proportions of baseline demographic characteristics and the Student’s *t* and Mann–Whitney *U* tests were used to compare continuous baseline characteristics with normal and non-normal distributions, respectively. The chi-squared test was also used to compare the proportions of patients who received appropriate colistin dosing or new recommendations. Logistic regression was used to perform a crude analysis of AKI incidence and its association with daily colistin dose. Multivariable logistic regression was used to adjust for potential confounders and identify predictors associated with the development of nephrotoxicity. The variables included in the model were selected based on prior knowledge, our data whenever feasible, and included age, BMI, baseline creatinine clearance, total daily colistin dose (mg/kg/day), use of vasopressor agents, use of diuretics, ICU admission, baseline serum albumin concentration, CCI, and APACHE II score. We assessed the model fitness using Hosmer Lemeshow’s goodness-of-fit test and the area under the receiver operator characteristic (ROC) curve. As some of these variables were missing in our data, we addressed the missingness of important variables which would impact the multivariate analysis. We used simple regression imputation [25,26] to handle missing baseline albumin values by developing a prediction model for missing baseline albumin values, which included the following predictors; gender, sex, ICU admission, CCI, and the total daily colistin dose (mg/kg/day). We used an indicator variable method [27] to handle missing non-applicable APACHE II scores of non-ICU patients. A two-sided *p*-value of <0.05 was considered to indicate statistical significance in all analyses. Data analysis was performed using STATA 14 (StataCorp LP, College Station, TX, USA).

### 2.7. Ethics

The study received approval from the institutional review board of the King Abdullah International Medical Research Center, Ministry of National Guard Health Affairs, Riyadh, Saudi Arabia (RJ16/048/J) in July 2016. The research was conducted according to the Declaration of Helsinki and national and institutional standards. We did not obtain informed consent from the participants given the retrospective nature of the study.

## 3. Results

### 3.1. Patients

In total, 467 patients were screened, of whom 198 met the inclusion criteria. The reasons for exclusion of the remaining 269 participants are demonstrated in Figure 1. Enrollment and screening flow diagram.

The mean age of the participants was 55.7 ± 19.36 years, 62% were men, the mean baseline creatinine clearance was 108.54 ± 46.26 mL/min, and the mean cumulative dose of colistin was 3.33 ± 0.99 mg/kg/day. Regarding baseline cultures, 60% (119/198) contained MDR organisms, 24.8% (49/198) contained no organisms, and 15.5% (30/198) contained non-MDR organisms. However, only 54% (107/198) of these isolates demonstrated sensitivity to colistin, 5.6% (11/198) were resistant to colistin, and 15.7% (31/198) did not have a reported sensitivity in the EHRs. The mean cumulative dose of colistin among patients who developed acute kidney injury was 195.54 ± 61.60 mg/day vs.198.33 ± 66.87 in those who did not develop AKI (*p* = 0.763). The baseline characteristics of the patients were stratified based on the development of AKI and are presented in Table 1.

### 3.2. Outcomes

#### 3.2.1. Primary Outcome

At the primary end point, AKI occurred in 88 of the 198 patients (44.4%; 95% CI: 37.4–51.7). Of those 88 patients, 39 (44.3%) had reversible AKI and their serum creatinine returned to baseline levels, whereas the serum creatinine of 44 (50%) patients did not revert to baseline levels and the recovery serum creatinine data of five patients were missing. The median time to AKI development was 6 days (IQR: 3.5–11 days).

#### 3.2.2. Secondary Outcomes

We noticed 38/198 missing values for serum albumin which had a normal distribution, so we used the following multiple imputation model to impute for missing serum albumin (Y).
Serum albumin (Y) = β_0_ + β_1_ Sex + β_2_ Age+ β_3_ Admission to ICU+ β^4^ Acute kidney injury + β_5_ CCI + β_6_ average colistin dose per weight per day(1)

β_0_: regression coefficient constant, β_1,_ β_2_ β_3_ β_4,_ β_5,_ β_6,_ β_7_ are the regression coefficients of the model variables (sex, age, admission to ICU, development of acute kidney injury, CCI, and average colistin dose per weight per day (mg/kg/day), respectively).

Subsequently, we replaced the missing values of the serum albumin at baseline with the results of the regression model for predicted albumin stated above.

As our study included 131 patients admitted to ICU, we had to impute for missing non-applicable APCAHE-II for non-ICU patients, so we used indicator variable method for a missing APCHE-II.

A univariate analysis revealed that age, BMI, baseline albumin concentration, baseline creatinine clearance, admission to ICU, CCI, APACHE II score, diuretic use, and vasopressor use were associated with colistin-induced AKI. Univariate analysis of the association between AKI development and cumulative daily colistin dosing (mg/kg/day) revealed no statistical significance. However, multivariate analysis revealed that, for each 1 mg/kg/day of colistin, the odds of developing AKI increased by ~1.6 times, which was statistically significant after adjustment for potential confounders as demonstrated in Table 2.

Furthermore, multivariate analysis demonstrated that only baseline serum albumin concentration, BMI, and CCI were the other predictors associated with AKI development (*p* = 0.004, 0.02 and 0.04, respectively), as reported in Table 2. The model demonstrated a Hosmer Lemeshow’s goodness-of-fit of 211.63, *p* = 0.10, and an area under the ROC curve of 0.80. Details for multivariate analysis are reported in Table 2.

More than 75% of the patients had Stage-1 or Stage-2 AKI. Table 3 illustrates the secondary outcomes, including AKI classifications of the stages.

Clinical cure occurred in 13% of only 70% (138/198) of our participants; we were unable to assess the clinical cure rate of the whole cohort at Day 7 because data were missing. Among the patients who did not achieve clinical cure, the infections of only 10 participants exhibited resistance to colistin. Other secondary outcomes, such as 30-day mortality and adherence to the dosing recommendations of the FDA [4], the EMA [6], and Garonzik et al. [8], are presented in Table 3.

## 4. Discussion

In this study, we observed a 44.4% incidence of colistin-associated AKI. Although this is consistent with the range previously reported for colistin-induced AKI, it is high given that the average daily colistin dose administered to our patients was lower than the recommended dose by FDA, EMA, and the most recent consensus guidelines for optimal use of the polymyxins published in 2019 [8,28,29,30,31]. In addition, a study reported an incidence of colistin-induced nephrotoxicity of 33.9%, despite the fact that the participants received average daily doses of 4.6 mg/kg/day (based on ideal body weight (IBW)) [32]. However, in that study, the Risk, Injury, Failure, Loss of Kidney Function, and End-Stage Kidney Disease criteria were used for the assessment of AKI and APACHE II scores and CCI were not reported, so we were unable to compare the severity of illness with that of our patients [32].

Another study evaluated patients receiving a similar dosing regimen but shorter courses of colistin (8 days; IQR: 5–14) compared with our cohort [33]. The study reported AKI in 29% of the participants at 7 days, despite the fact that 50% of the patients received a loading dose compared with none in our study [33]. It is unclear whether use of a loading dose and higher colistin maintenance doses are associated with an increased incidence of nephrotoxicity. The evidence supporting the use of a loading dose is lacking appropriate clinical outcomes; however, current data do not show a higher risk of nephrotoxicity with use of a loading dose [10,34]. Furthermore, the consensus guidelines for optimal use of the polymyxins published in 2019 acknowledged this gap of evidence, advocated the use of loading doses as the need to achieve rapid therapeutic colistin concentrations outweighs the possible risk of AKI development which may be associated with the use of loading doses [30].

In our study, after adjustment for potential confounders, the odds of nephrotoxicity increased by ~1.6 times for each 1 mg/kg/day of colistin. A previous study reported an association between cumulative colistin doses of ≥5 mg/day (based on IBW) and nephrotoxicity with an OR of 23.4 (95% CI: 5.3–103.6) [13]. Other predictors associated with AKI identified in our study included the severity of illness (characterized by a high CCI) and low baseline serum albumin concentration, consistent with previous literature [14,15,35,36,37]. Furthermore, almost all of our patients received nephrotoxic agents, which have been identified as predictors of nephrotoxicity [13,36].

Our study demonstrated an alarmingly low clinical cure rate, which should be interpreted with caution. First, our patients were underdosed with colistin according to the new consensus guidelines for the appropriate use of polymyxins in 2019 as per their baseline kidney function (mean baseline creatinine clearance was 108.54 ± 46.26 mL/min in our cohort) and none of our patients received a loading dose [30]. A previous study reported that a colistin dose of >4.4 mg/kg/day (based on IBW) was an independent predictor of global clinical cure rate and achieved a higher clinical cure rate at Day 7 (40%) than lower doses (22%; *p* = 0.03) [23]. Loading and higher maintenance doses of colistin may achieve a therapeutic concentration more rapidly. This new regimen may increase the higher clinical cure rate and reduce the duration of therapy [30,34]. Second, our cohort included severely ill patients, with an average APACHE II score of 20 (which is highly predictive of severity of illness) and a 40% mortality rate. Third, respiratory isolates were the most common sources of infection (47.5%); IV administration of colistin may have lower efficacy against respiratory infections because of its limited penetration into the pulmonary epithelial lining fluid [38]. Additionally, only 11.6% of our patients received both inhaled and IV colistin. This practice is not consistent with the 2016 Infectious Disease Society of America guidelines for the management of adults with hospital-acquired and ventilator-associated pneumonia caused by MDR and extensively drug-resistant strains of *Acinetobacter baumannii*, *Pseudomonas aeruginosa*, and Enterobacteriaceae published during the study period which recommended the use of adjunctive inhaled colistin in addition to IV colistin [39]. Furthermore, this recommendation is consistent with the most recent consensus guidelines for optimal use of the polymyxins published in 2019 [30]. Fourth, we cannot explain our low clinical cure rate in terms of the presence of colistin-resistant isolates (5%). However, the resistance rate may have been inaccurate because of use of the gradient test at our institution at the time of the study, which underestimates colistin MIC values and resistance as per the European Committee on Antimicrobial Susceptibility Testing (EUCAST) and Clinical and Laboratory Standards Institute (CLSI) statement [40]. Finally, our lab did not test for the genetic mutation for isolates of *Klebsiella pneumonia* at that time and ceftazidime-avibactam was not approved by Saudi Food and Drug Authorities until 2018.

Our study has several limitations. First, because of its retrospective design, we were unable to assess the appropriateness of indication for the use of colistin therapy given that it was started as per the discretion of the treating physician and there was no standard definition for an infection requiring the use of colistin. This may have influenced the clinical cure rates if colistin use was not the most suitable choice. Additionally, data were missing for variables such as baseline serum albumin concentration, vital signs and complete blood counts on day 7 post-treatment. These missing variables limited our ability to assess the clinical cure rate of the whole cohort. We believe that these data were missing at random or data on worst-case scenarios were missing completely at random. For example, all missing baseline serum albumin values were for non-ICU patients, who were expected to be haemodynamically stable and less critically ill than ICU patients. Although we attempted to overcome the missing values for serum albumin concentration by regression imputation, as it is an important variable which would impact the adjusted analysis, this may have resulted in residual confounding. Second, selection bias resulting from the inclusion of patients with complete data for the outcomes of interest may have caused us to overestimate the overall rate of AKI in our cohort. Third, we were unable to assess the association between various regulatory recommendations as we initially planned and the incidence of nephrotoxicity because of suboptimal dosing and the lack of loading doses as per the recommendations of Garonzik et al. [8] and the EMA [26]. However, our data can serve as historical comparison data to assess the implementation of the new consensus of polymyxin use as we integrated into our practice [28]. Fourth, our study has limited generalizability to ICU patients, who represent two-thirds of our cohort, and to patients with a high CCI. Finally, as we had 88 events of AKI, we had limitations in the number of variables that we could include in the model to avoid over fitness for the adjusted analysis.

Our study has several strengths. We adjusted for potential confounders based on prior knowledge and our data, and the model demonstrated a good fitness and a ROC curve. Additionally, we were able to follow a large number of patients during the course of colistin therapy through EHRs. Finally, our study used the total cumulative daily dose of colistin per kilogram of body weight as a novel, pragmatic way of expressing the association between colistin dosing and nephrotoxicity.

Use of colistin for the eradication of MDR organisms has multiple dimensions. Therefore, future studies should adopt a comprehensive approach that aims to improve clinical cure and minimize nephrotoxicity. Research should prospectively individualize colistin-dosing regimens, loading doses, and duration of treatment based on clinical infectious syndromes and therapeutic drug monitoring and aims to assess new dosing recommendations in a pragmatic setting [30].

## 5. Conclusions

In conclusion, our study demonstrated that colistin-associated nephrotoxicity was high in our population. Cumulative daily dose of colistin per body weight, BMI, serum albumin concentration, and severity of illness were independent predictors of nephrotoxicity. Future studies must assess new dosing of colistin regimens to provide insights on strategies that maximize target therapeutic outcomes and optimize patient safety.

## Figures and Tables

**Figure 1 antibiotics-09-00485-f001:**
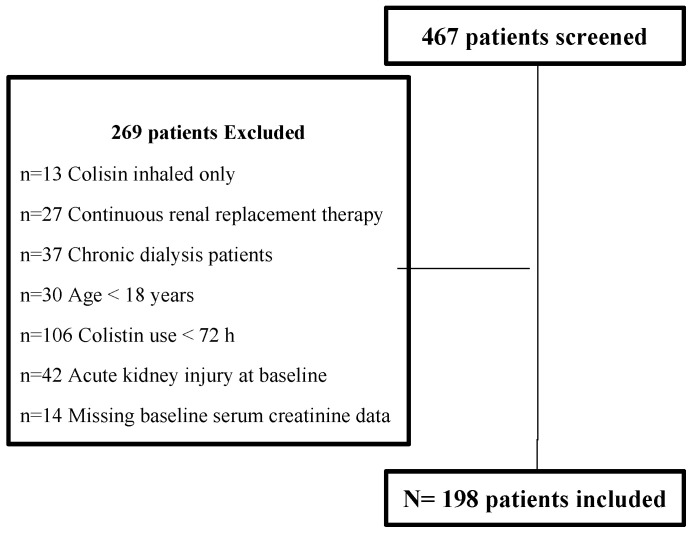
Enrollment and screening flow diagram.

**Table 1 antibiotics-09-00485-t001:** Baseline characteristics.

Baseline Characteristics	AKI ^1^ (*n* = 88)	No AKI ^1^ (*n* = 110)	*p*-Value ^2^
Age (years)	59.5 ± 17.7	52.62 ± 20.11	**0.011**
Sex (male)	50 (56.8)	72 (65.5)	0.214
Body mass index (kg/m^2^)	28 (24–34.2)	25.1 (21.5–28.7)	**0.003**
Baseline CrCL^3^ (mL/min)	99.79 ± 45.44	115.53 ± 45.91	**0.017**
ICU admission	65 (73.9)	66 (60)	**0.04**
ICU stay prior to colistin therapy (days)	3 (0–11)	6 (0–16)	0.079
Charlson Comorbidity Index (CCI)	5.5 (3–7)	3 (0–6)	**<0.001**
APACHE II score ^4^	22.93 ± 7.19	19.72 ± 6.91	**0.013**
Anaemia	84 (95.45)	102 (92.73)	0.424
Serum albumin concentration (g/L) ^5^	24.53 ± 4.9	27.85 ± 5.19	**<0.001**
**Colistin**			
Dosing body weight, DBW (kg) ^6^	58.64 ± 12.89	60.9 ± 13.45	0.392
Duration of colistin therapy (days)	11 (6–16.5)	12 (8–16)	0.203
Colistin dose (mg/kg/day) ^7^	3.29 ± 1	3.24 ± 0.97	0.702
**Causative organism**			0.182
*Acinetobacter baumannii*	28 (31.82)	43 (39.09)	-
*Klebsiella pneumoniae*	17 (19.32)	9 (8.2)	-
*Pseudomonas aeruginosa*	7 (8)	17 (15.5)	-
**Site of infection**			0.111
Respiratory	36 (40.9)	58 (52.7)	-
Blood	25 (28.4)	19 (17.3)	-
Urine	13 (14.8)	22 (20)	-
Other ^8^	14 (15.9)	11 (10)	-
**Antibiotic therapy**			0.667
Combined use of antimicrobial agents	76 (86.4)	94 (85.5)	-
Combined systemic, nebulizer use of colistin and other antibiotics	9 (10.2)	14 (12.7)	-
Other ^9^	3 (3.4)	2 (1.82)	
**Combined use with nephrotoxic agents**	87 (98.9)	106 (96.4)	0.265
ACEIs ^10^/ARBs ^11^	22 (25)	21 (19.1)	0.316
Intravenous contrast media	39 (44.3)	43 (39.1)	0.458
Diuretics	67 (60.9)	72 (81.8)	**0.001**
Vasopressors	59 (67.1)	46 (41.8)	**0.001**
Aminoglycosides	15 (17.1)	11 (10)	0.145
Amphotericin	10 (11.4)	2 (1.82)	0.005
Vancomycin	78 (88.6)	86 (78.2)	0.053
Acyclovir	11 (12.5)	13 (11.8)	0.884
Non-steroidal anti-inflammatory drugs	2 (2.3)	4 (3.6)	0.578

^1^ Data are reported as means ± standard deviation for continuous variables or medians (interquartile ranges) and numbers (%) as appropriate. ^2^
*p*-value based on the chi-squared test or Fisher’s exact test or the *t*-test and Mann–Whitney *U* test as appropriate, with *p* < 0.05 indicating significance (in bold). ^3^ Baseline CrCL (Creatinine Clearance) based on the Cockcroft–Gault equation. ^4^ APACHE II score for the prediction of ICU mortality calculated within 24 h of ICU admission, a value missing for all non-ICU patients. ^5^ Serum albumin concentration at baseline, including imputed values for 38 patients. ^6^ DBW based on actual body weight or ideal body weight (IBW) if actual body weight is ≥130% of the IBW. ^7^ Colistin dosing per body weight. ^8^ Other infections, including complicated skin and soft tissue infections, intra-abdominal infections, and catheter tip infections. ^9^ Other combinations of systemic and inhaled colistin and monotherapy. ^10^ ACEIs: angiotensin-converting enzyme inhibitors. ^11^ ARBs: angiotensin receptor blockers.

**Table 2 antibiotics-09-00485-t002:** Univariate and multivariate analysis for predictors of acute kidney injury (AKI).

Variables	Univariate Analysis Odd Ratio (OR), 95% CI, *p*-Value (*p*)	Multivariate Analysis OR, 95% CI, *p*-Value
Colistin dose (mg/kg/day)	1.09, 95% CI (0.83–1.46), *p* = 0.52	OR: 1.57, 95% CI (1.08–2.30), *p* = 0.02 ^1^
Age (years)	1.02, 95% CI (1.003–1.035), *p* = 0.01	0.98, 95% CI (0.96–1.01), *p* = 0.26
Sex (female)	1.44, 95% CI (0.81–2.56), *p* = 0.215	NA ^2^
BMI (kg/m^2^)	1.07, 95% CI (1.03–1.12), *p* < 0.001	1.06, 95% CI (1.01–1.11), *p* = 0.02
Creatinine clearance (m//min)	0.99, 95% CI (0.986–0.998), *p* = 0.02	0.99, 95% CI (0.98–1), *p* = 0.09
Admission to ICU	1.88, 95% CI (1.02–3.47), *p* = 0.042	0.25, 95% CI (0.03–1.90), *p* = 0.18
Hospitalization in ICU prior to colistin use (days)	0.99, 95% CI (0.99–1.004), *p* = 0.24	NA ^2^
Charlson Comorbidity Index	1.20, 95% CI (1.09–1.33), *p* = <0.001	1.16, 95% CI (1–1.34), *p* = 0.04
APCHAE-II score	1.04, 95% CI (1.02–1.07), *p* = <0.001	1.03, 95% CI (0.97–1.10), *p* = 0.31
Serum albumin (g/L)	0.87, 95% CI (0.81–0.93), *p* = <0.001	0.89, 95% CI (0.83–0.96), *p* = 0.004
Use of nephrotoxic agent	3.28, 95% CI (0.36–29.91), *p* = 0.29	NA ^3^
Diuretic use	2.89, 95% CI (1.49–5.60), *p* = 0.002	1.01, 95% CI (0.43–2.36), *p* = 0.99
Vasopressor use	2.83, 95% CI (1.58–5.08), *p* < 0.001	2.1, 95% CI (0.90–4.91), *p* = 0.09

^1^ Adjusted for age (years), BMI (kg/m^2^), baseline creatinine clearance (mL/min), admission to ICU, Charleston Comorbidity Index, baseline APACHE II score, serum albumin concentration (g/L), diuretic and vasopressor use. ^2^ NA: Not applicable as sex and hospitalization in ICU prior to colistin use (days) were not significantly associated with AKI in the univariate analysis so were excluded from the multivariate analysis. ^3^ NA: Not applicable as concomitant use of nephrotoxic medications with colistin was dropped from the model because almost all patients received one nephrotoxic agent so it cannot be a predictor.

**Table 3 antibiotics-09-00485-t003:** Secondary end points.

Secondary End Points	AKI 88 (44.4) *n* (%)	No AKI 110 (55.5) *n* (%)	*p*-Value
AKI classifications ^1^			
Stage I	29 (33)	-	-
Stage II	39 (44.3)	-	-
Stage III	20 (22.7)	-	-
Clinical cure ^2^	5 (5.7)	13 (11.8)	0.136
Mortality at 30 days	42 (47.7)	24 (21.8)	<0.001
Dosing appropriateness			
FDA ^3^	63 (72)	74 (67.3)	0.513
EMA			
Loading dose	None	None	-
Maintenance dose	5 (5.7)	10 (9.1)	0.368
Garonzik et al. [8]			
Loading dose	None	None	-
Maintenance dose	5 (5.7)	0 (0)	-

^1^ Acute kidney injury (AKI) classification according to the Kidney International global outcomes criteria for the staging of AKI. ^2^ Clinical cure: 60 patients were missing (comprising 40 patients who received colistin for <7 days and 20 patients missing White Blood Cells data on Day 7). ^3^ Calculated based on FDA recommendations and categories of creatinine clearance at baseline.

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
