# Peer review of "Impact of Colistin Dosing on the Incidence of Nephrotoxicity in a Tertiary Care Hospital in Saudi Arabia"

_antibiotics, 2020, doi:10.3390/antibiotics9080485_

Round 1

Reviewer 1 Report

In this manuscript, Reem Almutairy et al investigated the colistin using in the Saudi Arabia and found the inappropriate use can lead to high incidence of colistin-associated AKI. Also, they found independent predictors of nephrotoxicity induced by colistin including daily dosage, body mass index, serum albumin level and the severity of illness. It’s well designed and can be accepted with minor revision as to typos as “the its” on line 75, page 2.

Author Response

Reviewer 1

Comments and Suggestions for Authors

In this manuscript, Reem Almutairy et al investigated the colistin using in the Saudi Arabia and found the inappropriate use can lead to high incidence of colistin-associated AKI. Also, they found independent predictors of nephrotoxicity induced by colistin including daily dosage, body mass index, serum albumin level and the severity of illness. It’s well designed and can be accepted with minor revision as to typos as “the its” on line 75, page 2.

In this manuscript, Reem Almutairy et al investigated the colistin using in the Saudi Arabia and found the inappropriate use can lead to high incidence of colistin-associated AKI. Also, they found independent predictors of nephrotoxicity induced by colistin including daily dosage, body mass index, serum albumin level and the severity of illness. It’s well designed and can be accepted with minor revision as to typos as “the its” on line 75, page 2.

Thank you for your feedback and review for the manuscript

 The typo changed has been corrected in the original manuscript with track changes ( we removed the word ‘the” )

Reviewer 2 Report

Antibiotics: Manuscript ID#:875310

Tile - Impact of colistin Dosing on the Incidence of Nephrotoxicity in a Tertiary Care Hospital in Saudi Arabia.

General – The manuscript is well written, easy and clear to follow.  The only major issues noticed as I reviewed the manuscript is inconsistency in the abbreviations and one or two areas of clarification.  Also, few references may not be appropriately placed.    

ABSTRACT (Page #1)

Lines #20 – Line #34

Per Authors guidelines, abbreviations are allowed in the abstract, main text, tables, figures, etc. and be used consistently in thereafter.  Few can be adjusted in this section.

Line #22 – Consider inserting the “IV” abbreviation after the intravenous since it has been repeated later.

Line #25 – Insert the “FDA” abbreviation just after.

Line #26 – Insert the “EMA” abbreviation here as well for consistency. 

Line #30 – Consider inserting “BMI” abbreviation after the body mass index.

Line #31 – Line #32 - The sentence may need an additional punctuation or wording.  Not sure if it is supposed to read “None of the patients received a loading dose; however, FDA recommended dosing was achieved in 70.2%???  Please review and clarify.   

Line #33 – Use the BMI abbreviation here since it was just defined in line #30.

INTRODUCTION

Line #73 – Should the references be listed here as well for Garonzik and EMA?

MATERIALS and METHODS

Study Participants

Line #88 – Just use the “IV” abbreviation since it was already defined and abbreviated in the abstract.

Line #89 – AKI has been abbreviated multiple times already and delete the “acute kidney injury” and just use AKI.

Statistical Analysis

Line #129 – IQR abbreviated here for the first time; consider spelling it out first.

Line #143 – ROC is Receiver Operator characteristic (ROC) curve.

Study Outcomes

Line # 101 and Line #102 – Is this the same guideline since the reference is the same?  If so, then just use the KDIGO abbreviation.

Statistical Analysis

Line #129 – IQR (interquartile range) is seen for the first time and is abbreviated.  Please define first.

Line #143 – ROC is receiver operating characteristic (ROC) curve.  Please correct the wording for completeness.

Secondary Outcomes

Line #208 – Abbreviate “BMI”.

Line #225 - Abbreviate “BMI”.

Line #227 – “ROC” has been abbreviated (line #143), so just use ROC curve.

DISCUSSION

Line #271 – Line #297 - Not sure if I have seen firstly, thirdly, fourthly.  It should just be plain first, second, third, fourth, etc…

Line #288 – Should Enterobacteriaceae also be italicized? Enterobacteriaceae

Line #298 – Line #319 – Same as above, first, second, third, fourth, and finally (is good/appropriate).

CONCLUSIONS

Line #333 – Abbreviate “BMI”

TABLES

Nice work, no comments.

FIGURE(s)

The number of patients excluded does not add up to 269, it is only 213; is there a group missing?

May also consider listing them as: n=13, n=27, n=37, etc. 

REFERENCES

Line #53 and Line #54 – Here you have FDA and EMA update on dosing recommendations and the reference numbers are [3 & 6].   Is this just based on the package insert update?

Reference #4 – is listed as the FDA: but, then on page 2, Line #57 you have FDA recommendations [7].  Please verify.

Author Response

Reviewer 2

Comments and Suggestions for Authors

Antibiotics: Manuscript ID#:875310

Tile - Impact of colistin Dosing on the Incidence of Nephrotoxicity in a Tertiary Care Hospital in Saudi Arabia.

General – The manuscript is well written, easy and clear to follow.  The only major issues noticed as I reviewed the manuscript is inconsistency in the abbreviations and one or two areas of clarification.  Also, few references may not be appropriately placed.    

Thank you for your feedback and review of the manuscript

Kindly find below itemized response to the comments in blue

ABSTRACT (Page #1)

Lines #20 – Line #34

Per Authors guidelines, abbreviations are allowed in the abstract, main text, tables, figures, etc. and be used consistently in thereafter.  Few can be adjusted in this section.

  • Line #22 – Consider inserting the “IV” abbreviation after the intravenous since it has been repeated later.
  • Line #25 – Insert the “FDA” abbreviation just after.
  • Line #26 – Insert the “EMA” abbreviation here as well for consistency. 
  • Line #30 – Consider inserting “BMI” abbreviation after the body mass index.

All these suggested requested abbreviations are included in the abstract, It was a bit challenging so we have done some tiny changes in the abstract to fit into the requirements of the journal of 200 words

  • Line #31 – Line #32 - The sentence may need an additional punctuation or wording.  Not sure if it is supposed to read “None of the patients received a loading dose; however, FDA recommended dosing was achieved in 70.2%???  Please review and clarify.   

We changed as suggested:” None of the patients received loading doses, however FDA recommended dosing was achieved in 70.2% and the clinical cure rate was 13%”

  • Line #33 – Use the BMI abbreviation here since it was just defined in line #30.

Done at page 1

INTRODUCTION

  • Line #73 – Should the references be listed here as well for Garonzik and EMA?

This sentence point to factors for nephrotoxicity, however, the EMA and Garonzik references refer for dosing

Several risk factors can increase a patient's risk of developing colistin-induced nephrotoxicity, including chronic kidney disease, diabetes mellitus, cancer, anaemia and exposure to multiple events harmful to the kidneys (e.g., critical illness, sepsis, shock, burns, trauma, major surgery and nephrotoxin exposure) [15, 16]. However, colistin-induced nephrotoxicity is reversible and rarely results in permanent kidney damage [17, 18]. The risk of nephrotoxicity must be weighed against the consequences of inadequately treating a life-threatening infection [17, 18].

MATERIALS and METHODS

Study Participants

  • Line #88 – Just use the “IV” abbreviation since it was already defined and abbreviated in the abstract.
  • Line #89 – AKI has been abbreviated multiple times already and delete the “acute kidney injury” and just use AKI.

All the suggested changes and abbreviations are included in the text: line 90 & line 99 at page 2 &3 respectively

Statistical Analysis

  • Line #129 – IQR abbreviated here for the first time; consider spelling it out first.
  • Line #143 – ROC is Receiver Operator characteristic (ROC) curve.

All the suggested changes and abbreviations are included in the text: line 138 &line 164 at page 3 & 4 respectively

Study Outcomes

  • Line # 101 and Line #102 – Is this the same guideline since the reference is the same?  If so, then just use the KDIGO abbreviation.

Done at page 3, line 108

Statistical Analysis

  • Line #129 – IQR (interquartile range) is seen for the first time and is abbreviated.  Please define first.
  • Line #143 – ROC is receiver operating characteristic (ROC) curve.  Please correct the wording for completeness.

All the suggested changes and abbreviations are included in the text: line 138 &line 164 at page 3 & 4 respectively.

Secondary Outcomes

  • Line #208 – Abbreviate “BMI”.
  • Line #225 - Abbreviate “BMI”.

I changed all body mass index in the manuscript to BMI after first defining in the abstract as suggested at line 31, 161, 237, table 2, 249 , 255 and 453

  • Line #227 – “ROC” has been abbreviated (line #143), so just use ROC curve.

Changed at line 260,

DISCUSSION

  • Line #271 – Line #297 - Not sure if I have seen firstly, thirdly, fourthly.  It should just be plain first, second, third, fourth, etc…

Changed as suggested at Line 310 – 333- page 9

  • Line #288 – Should Enterobacteriaceae also be italicized? Enterobacteriaceae

Done, line 326, italicized

  • Line #298 – Line #319 – Same as above, first, second, third, fourth, and finally (is good/appropriate).

All the recommendations are carried out on the revised manuscript from line 336- 439 (page 10)

CONCLUSIONS

  • Line #333 – Abbreviate “BMI”

Done

TABLES

  • Nice work, no comments.

Thank you

FIGURE(s)

  • The number of patients excluded does not add up to 269, it is only 213; is there a group missing?
  • Thank you for pointing to this, it seems that the box for exclusion patients was disfigured when we copied to the journal ‘s template , we expanded the box and the two categories for acute kidney injury at baseline (42) and missing values (14 ) for baseline serum creatinine are clearly shown now .
  • All exclusion reasons sum up to 269 in figure 1 in the revised version of the manuscript
  • May also consider listing them as: n=13, n=27, n=37, etc. 

Yes, thank you, we included n for each category of the exclusion and added N for the total number of included participants.

REFERENCES

  • Line #53 and Line #54 – Here you have FDA and EMA update on dosing recommendations and the reference numbers are [3 & 6].   Is this just based on the package insert update?

Yes, it is based on the package insert update in the year of 2017

  • Reference #4 – is listed as the FDA: but, then on page 2, Line #57 you have FDA recommendations [7].  Please verify.

Thank you. I would like to point to three different references included in the text:

  1. Background: Page 1, Line 44, refers to the FDA leaflet at 2006
    1. Food and Drug Administration. Coly-Mycin® M Parenteral (Colistimethate for Injection,USP).https://www.accessdata.fda.gov/drugsatfda_docs/label/2009/050108s026lbl.pdf. Accessed on July 21st, 2020.
  2. Background: Page 2, line 62 refers to Nation el al study [7] Which demonstrates that the European dosing achieve target steady state concentration than FDA dosing. However, I added reference 4 which present FDA recommendations to the citation of this sentence to make it clear which two references we are comparing against each other
    • “A recent study that compared both dosing regimens found that the EMA regimen, which includes higher overall maintenance doses, is more likely to achieve the target steady state concentration of colistin than the FDA recommendations [4,7].”
    • Nation RL, Garonzik SM, Li J, Thamlikitkul V, Giamarellos-Bourboulis EJ, Paterson DL, et al. Updated US and European Dose Recommendations for Intravenous Colistin: How Do They Perform? Clin Infect Dis. 2016;62(5):552-8
  3. Discussion section:
  • Page 8, line 274, the FDA reference is pointing to the updated changes in the prescribing of FDA label in 2017 (Different reference ID number for the document (Reference ID: 4084464) à reference 29
  • Food and Drug Administration. Coly-Mycin® M Parenteral (Colistimethate for Injection, USP. https://www.accessdata.fda.gov/drugsatfda_docs/label/2017/050108s033lbpdf. Accessed on July 29th, 2020

Reviewer 3 Report

Major comments:

Line 170: “24.8% (49/198) contained no organisms”.  When were the patients tested for infection, before prescribing colistin or after?  Since 25% is a large number, the obvious question to ask is about the standard practice at the hospital for prescribing colistin. 

Line 170: “(30/198) contained susceptible organisms”.  Susceptible to what? To colistin?  Then why in the next line it states “(107/198) of the isolates demonstrated sensitivity to colistin”?  Why are the two numbers different?  Is there a difference between susceptible and sensitive? 

Line 172: “only 54% (107/198) of the isolates demonstrated sensitivity to colistin; 15.7% (31/198) did not have a reported sensitivity and 5.6% (11/198) were resistant to colistin.”  In the previous sentence the authors say that 49 patients did not have any organism (infection?).  In that case, how can these samples be classified as resistant or sensitive to colistin?  If there is no organism then what is resistant or sensitive? 

Line 171: “15.7% (31/198) did not have a reported sensitivity”.  It is not clear what this means.  This is a study of patients undergoing colistin therapy.  How is it possible that 31 of them did not have reported sensitivity and also were not resistant?  What other possibility is there besides being sensitive or resistant?

This whole paragraph makes no sense to me. 

Figure 1.  The purpose of Figure 1 is not clear.  The information in the figure can be simply stated in a few sentences as text.  In fact, this has been actually been stated in the text (lines 92-95) but unfortunately the information in the figure does not agree with that in the text.  For example, in the figure there is no mention of pregnant patients or patients who received more than one course of colistin. 

Table 1: It is not clear what these numbers are for?  What are the numbers in parentheses? It is not mentioned but I am guessing that in some cases the numbers represent the heading on the left for that row while in some cases they represent the heading on the top of the column and in some it is not clear at all what they represent.  I am also guessing that in some cases the numbers in the parenthesis represent a percentage while in some others they represent a range of measured units.  There is no consistency at all in this table. 

This table is making no sense to me.

Table 2: Same comments as for Table 1.  What do these numbers represent?  For example, in the second row first column it is written “Age (years)”, which should mean that the numbers on the right are age in years, which they obviously are not.  Same comment for all the lines.

Table 3: Column 2 last two rows: Both numbers are the same (5) but the percentages are different (5.7 and 6.8).  Why?

Table 3: for both columns 2 and 3, the last three rows, the numbers don’t add up to 100 %.  Why?

Minor comments:

Line 43: Add space after “mortality”

Line 44 and other places: Introduce abbreviations when they are first mentioned or somewhere in the first page (depending on journal policy).  Examples: FDA, EMA, AKI (first mentioned in line 74 but introduced in line 89), IQR (line 129), CI (line 131)

Line 90: Change “as defined as” to “as defined by”.

Line 114: Change “risk of mortality risk” to “risk of mortality”.

Line 143: Change “(ROC) curve” to “curve (ROC)”.

Line 190: Change “Thirty-nine patients (44.3%)” to “Of the 88 patients 39 (44.3%)”.

Line 197-202: “multiple imputation model”.  Where is this formula taken from?  Give a reference.  If the authors have designed the formula, more explanation is needed.

Line 206: “indicator variable method”: Need reference.

Line 265: “In our study, after adjustment for potential confounders, the odds of nephrotoxicity increased by ~ 1.6 times for each 1 mg/kg/day of colistin.”  Refer to which Table and which line shows this result. 

Line 267: “Other predictors associated with AKI identified in our study included the severity of illness (characterised by a high CCI) and low baseline serum albumin concentration”.  Refer to which Table and which line shows this result. 

Author Response

Thank you for your review and feedback , kindly find attached our reply

Round 2

Reviewer 3 Report

In my previous review, I tried to bring to the attention of the authors some parts of the paper that readers may find confusing.  The authors have now answered all my questions and made the necessary changes making the manuscript much better.